# Postoperative Cerebral Venous Sinus Thrombosis Following a Retrosigmoid Craniotomy—A Clinical and Radiological Analysis

**DOI:** 10.3390/brainsci13071039

**Published:** 2023-07-07

**Authors:** Lukasz Przepiorka, Katarzyna Wójtowicz, Katarzyna Camlet, Jan Jankowski, Sławomir Kujawski, Laretta Grabowska-Derlatka, Andrzej Marchel, Przemysław Kunert

**Affiliations:** 1Department of Neurosurgery, Medical University of Warsaw, 02-097 Warsaw, Poland; 2Department of Exercise Physiology and Functional Anatomy, Ludwik Rydygier Collegium Medicum in Bydgoszcz Nicolaus Copernicus University in Toruń, 85-077 Bydgoszcz, Poland; 3Second Department of Radiology, Medical University of Warsaw, 02-097 Warsaw, Poland

**Keywords:** dural, sinus, thrombosis, craniotomy, cerebellopontine, tumor

## Abstract

Postoperative cerebral venous sinus thrombosis (CVST) is a rare complication of the retrosigmoid approach. To address the lack of literature, we performed a retrospective analysis. The thromboses were divided into those demonstrating radiological (rCVST) and clinical (cCVST) features, the latter diagnosed during hospitalization. We identified the former by a lack of contrast in the sigmoid (SS) or transverse sinuses (TS), and evaluated the closest distance from the craniotomy to quantify sinus exposure. We included 130 patients (males: 52, females: 78) with a median age of 46.0. They had rCVST in 46.9% of cases, most often in the TS (65.6%), and cCVST in 3.1% of cases. Distances to the sinuses were not different regarding the presence of cCVST (*p* = 0.32 and *p* = 0.72). The distance to the SS was not different regarding rCVST (*p* = 0.13). However, lower exposure of the TS correlated with a lower incidence of rCVST (*p* = 0.009). When surgery was performed on the side of the dominant sinuses, rCVSTs were more frequent (*p* = 0.042). None of the other examined factors were related to rCVST or cCVST. Surgery on the side of the dominant sinus, and the exposing of them, seems to be related with rCVST. Further prospective studies are needed to identify the risk factors and determine the best management.

## 1. Introduction

The retrosigmoid approach (RSA) is one of the most common neurosurgical approaches to the posterolateral skull base [1,2,3]. It provides exposure of the cerebellopontine angle region (CPA) and its surroundings, from Meckel’s cave to the jugular foramen [4,5,6]. Postoperative cerebral venous sinus thrombosis (CVST) is a rare but known complication following RSA [7].

There is a paucity of literature regarding CVSTs. Postoperative CVSTs have a wide range of presentation, from asymptomatic to death, and management is difficult. Treatment is inferred from a spontaneously occurring CVST, after which aggressive anticoagulation is immediately initiated [8]. Such management in the early postoperative period may increase the risk of an intracranial hemorrhage. Further, it is unclear whether any treatment should be started in asymptomatic thrombosis.

To address the above, we performed radiological and clinical analyses to report our experience with CVSTs after RSA. We started with categorizing the CVSTs. In the radiological part of the study, we looked for the radiologic features of CVSTs and evaluated the extent of the bony opening in relation to the sinuses. In the clinical part, we looked for the risk factors for, and clinical manifestations of, CVSTs (cCVSTs).

The aim of our study was to assess the frequency of radiologic features of CVSTs and cCVSTS. We assessed the relationships between clinical symptoms and the radiologic features of CVSTs. We looked for CVST risk factors, with particular emphasis on evaluating the relevance of transverse and sigmoid sinus exposures during bony openings.

## 2. Materials and Methods

### 2.1. Study Design

We designed this study as a retrospective, single-center evaluation of radiological and clinical data. Radiological data consisted of contrast-enhanced computed tomographic head scans performed routinely after elective RSAs, usually between postoperative days 3 and 7. Clinical data incorporated medical charts from RSA hospitalization.

We included adults undergoing elective RSAs, from 2016 to 2021, for which postoperative CT and medical charts were available. We could not reconstruct and evaluate studies prior to 2016 because of their insufficient quality. A CVST prior to surgery was an exclusion criterion. Patients were identified by reviewing all operative reports in our institution during the study period.

### 2.2. CVST Classification

CVSTs were divided into those demonstrating radiological features of CVSTs (rCVSTs, recognized retrospectively, as described in detail below) and cCVSTs, for which a clinical diagnosis of CVST had been established during hospitalization. 

### 2.3. Radiological Analysis

CT studies were reviewed by two teams, consisting of two neurosurgeons (attending and resident) and two medical students, all of whom were previously instructed by the attending neuroradiologist. We performed evaluations with GE software on radiological stations with diagnostic monitors. CT scans were evaluated for features of rCVST in the sigmoid sinus, transverse sinus, and their junction (each treated as a separate entity) on the ipsilateral side to the RSA. Recognizing rCVSTs required the visualization of a lack of contrast in the sinuses in postoperative CT scans. We considered intraluminal thrombi that partially or completely filled the sinuses as rCVST. Arachnoid (Pacchionian) granulations [9,10] were distinguished from rCVSTs by their distinctive, regular shape, in contrast to rCVSTs, which are irregular and often larger. We analyzed CT scans without any previous knowledge of the medical history of a patient or radiological report. We then reviewed the radiological reports of patients with rCVSTs, if these included descriptions of sinus thromboses. In addition, we evaluated the closest distance from the edge of the craniotomy to the sinuses on postoperative CT scans. It was measured and presented so that a positive value expressed the closest distance between the edge of the craniotomy and the sinus, whereas a negative value represented the magnitude of overlap of the craniotomy with the sinus (Figure 1). We performed such measurements for sigmoid and transverse sinus using axial and coronal scans, respectively. These continuous measurements were subsequently translated into qualitative evaluations of sinus exposure (which corresponded to all nonnegative measurements) and non-exposure of the sinus.

Furthermore, we evaluated the dominant side of the transverse-sigmoid sinuses, presence of intraparenchymal hematoma, and other cerebral sinus thrombosis (e.g., superior sagittal sinus).

### 2.4. Clinical Analysis

We reviewed the medical charts of analyzed patients to evaluate perioperative-, tumor-, procedure- and patient-related factors. Other venous thromboses (deep venous thrombosis, pulmonary embolism, etc.) were not studied.

### 2.5. Statistical Analysis

We used the means of two measurements for continuous variables (e.g., distance to the sinus). In case of any disagreement over qualitative variables (e.g., the presence of a rCVST), a final decision was made after discussion with a senior neuroradiologist. We additionally tried to find a threshold for the distance from the craniotomy to the sinus (Figure 1) in which the risk of CVST rises significantly. We examined the association between qualitative variables using the chi-squared test or Fisher’s exact test. The Kendall rank correlation coefficient was used to examine the relationship between ordinal and qualitative variables. Between-group differences in quantitative variables were examined using an independent samples t-test, Welch’s *t*-test, or Mann–Whitney U test, depending on whether the assumptions were met. ROC analysis was performed using the “MaxSpSe” method in the R package ‘OptimalCutpoints’ (R packages retrieved from MRAN snapshot 1 January 2022) [11,12].

## 3. Results

### 3.1. Patient Demographics

We included 130 patients (males: 52, females: 78) with a median age of 46.0 (Appendix A). Table 1 presents the radiological analysis and Table 2 presents the clinical analysis.

### 3.2. Radiological and Clinical Results

We recognized rCVSTs in 46.9% (61/130) of the study population, and most often in the transverse sinus.

### 3.3. Patients with cCVST

In our study group (*n* = 130) there were 4 cases of cCVSTs (Appendix A). 

One of the four patients with a cCVST died: a 41-year-old female was discharged home after RSA in a good condition. She was readmitted in a critical condition two days after discharge with a superior sagittal sinus thrombosis. She underwent unsuccessful endovascular treatment before dying in the intensive care unit. Of note, we did not recognize rCVST in the transverse or sigmoid sinus in this patient in our retrospective radiological analysis.

The second patient, a 46-year-old female, reported headaches on postoperative day six, before suddenly deteriorating neurologically. Her CT scan revealed a CVST with an intraparenchymal hematoma in the temporal lobe; she underwent an emergency craniotomy with hematoma removal and temporal lobe resection. Immediately after the surgery, she started improving and was discharged home, able to walk unassisted.

A 66-year-old male was diagnosed with a small intraparenchymal hemorrhage in the basal part of the temporal lobe, in his routine contrast-enhanced CT scan. A CVST diagnosis was confirmed using MRI, and this was followed by medical treatment. The patient was subsequently transferred to the neurology department for rehabilitation.

The fourth patient, a 42-year-old female, presented with aphasia on postoperative day two. After confirming a CVST with MRI, she was treated medically, then her symptoms resolved.

Half (2/4) of patients with cCVST had documented postoperative headaches refractory to regular medical treatment. However, severe postoperative headaches developed 9.2% of the whole study group (12/130). A comparison of patients with and without rCVST revealed no significant difference in postoperative headache frequencies (6/61, 9.8% and 6/69, 8.7%, respectively). 

In our retrospective radiologic studies, we identified 3 rCVSTs in 4 cases of cCVSTs. The remaining patient had superior sagittal sinus thrombosis, but no rCVST in their transverse/sigmoid sinus—which was defined as a rCVST in our study.

### 3.4. Radiological and Clinical Analysis

For the whole study population, median distances to the sigmoid and transverse sinuses were −1.75 mm and −6.60 mm, respectively; 60.77% of the sigmoid sinuses and 83.1% of the transverse sinuses were at least partially exposed. The right-sided sinuses were dominant in 72.3% of the cases.

Distances to the sigmoid and transverse sinuses were not significantly different between patients with and without cCVST (*p* = 0.340 and *p* =0.707, respectively, Student’s *t*-test). The distance to the sigmoid sinus was not significantly different in the group with rCVST in comparison to the group without rCVST (*p* = 0.125, Mann–Whitney U test). However, reduced exposure of the transverse sinus correlated with reduced frequency of rCVST (*p* = 0.028, Student’s *t*-test).

When we analyzed the distance to the sigmoid sinus and rCVST in the sigmoid sinus only, there was a significant difference: a rCVST in the sigmoid sinus was related to significantly larger exposure of the sigmoid sinus (*p* = 0.04, Mann–Whitney U test).

A similar analysis for the transverse sinus did not reveal such a correlation (*p* = 0.209, Welch’s test). A receiver operating characteristic curve analysis for the discrimination threshold for rCVST presence revealed that exposing a transverse sinus by over 6.55 mm increases the risk of rCVST (sensitivity 39%, specificity 41%, positive predictive value 37%, negative predictive value 43%, area under curve 0.367 (95% confidence interval = 0.269; 0.464), Figure 2).

We compared the exposing of each of the sinuses in a qualitative manner (exposed versus not exposed), and it correlated neither with rCVST nor cCVST (both *p* = 1, Fisher’s exact test for exposing sigmoid and transverse sinus and cCVST; and *p* = 0.16 χ^2^ test and *p* = 0.35, Fisher’s exact test for exposing sigmoid and transverse sinus and rCVST). Similarly, exposing both sinuses did not correlate with rCVST or cCVST (*p* = 0.460 and *p* = 0.1, Fisher’s exact test, respectively).

The side of sinus dominance did not correlate with cCVST or rCVST. However, when the surgery was on the side of the dominant sinus, a rCVST occurred more frequently (*p* = 0.042, χ^2^ test). We did not find such a correlation for cCVST (*p* = 0.631, Fisher’s exact test).

None of the remaining examined factors correlated with cCVST or rCVST, except for an oncologic past medical history being more frequent among the cCVST group (*p* = 0.03, Fisher’s exact test). In binary analyses, the presence of any risk factor (compared to none) did not correlate with cCVST (*p* = 0.609, Fisher’s exact test), nor with rCVST (*p* = 0.486, χ^2^ test). In addition, when the number of risk factors was treated as an ordinal variable, there was no correlation with cCVST (*p* = 0.233, Kendall rank correlation coefficient) or with rCVST (*p* = 0.437, Kendall rank correlation coefficient).

Intraoperative injuries of the sinuses did not correlate with rCVST or cCVST (*p* = 0.705, Fisher’s exact test; *p* = 1, Fisher’s exact test, respectively). Analysis of the radiological and clinical risk factors is summarized in Table 3.

## 4. Discussion

In this study, we retrospectively analyzed CVST after RSA. To our great surprise, an rCVST occurred after nearly every second (46.9%) surgery. Yet only a small portion of these cases (4.9%) bore clinical significance. Except for the occurrence, there is a handful of other topics arising from the results that will be explored below.

### 4.1. Mechanisms and Risk Factors

Possible risk factors of CVST after RSA have already been discussed in the literature [13,14], which can be divided into patient- and surgery-related. Patients at an increased risk of CVST (e.g., with the factor V Leiden mutation) should be identified prior to surgery. Outside of medical conditions, Gerges et al., in an anatomic study, found that acute petrous angle and shorter IAC to sinus distance were associated with CVST [15]. CVST prevention is yet to be explored, but clinicians might consider perioperative IV fluids and early anticoagulation postoperatively. In our series, only oncological past medical history correlated with cCVST, but this may be a statistical error caused by a small sample size, given the statistical test’s low power.

We should also recognize surgery-related risk factors for CVST, and tailor our surgical approaches to avoid them. Intraoperative maneuvers that are speculated to increase the risk of CVST include direct and indirect sinus injuries, along with the migration of bone wax used on emissary veins [13]. Interestingly, an intraoperative injury of the sinus in our material was not associated with any type of CVST, as reported by Apra et al. [16]. The importance of the surgical position for RSA is yet to be evaluated; we routinely choose a supine position with the head turned and fixed in a skull clamp.

In our institution, the craniotomy technique is focused on minimal (ideally no) exposure of the sigmoid and transverse sinuses. For this, we evaluate preoperative CT scans, use anatomical landmarks, and place the initial burr hole just below the expected junction of the sinuses [1]. As a rule, we do not use neuronavigation for RSA [17].

In case of a sinus injury, our strategy is tailored to the extent of bleeding. Just exposing the sinus, even partially, is followed by keeping it moist and applying a layer of an absorbable hemostat (e.g., Surigcel^®^) on the surface or just the patties. Minor bleedings are dealt with similarly. More serious bleedings are most often successfully stopped with a fibrin sealant patch (e.g., TachoSil^®^). A direct sinus injury is sutured.

When designing our study, we conjectured that exposing sinuses—even partially—may cause CVSTs. To some degree, this has been confirmed, but this requires analysis with a larger sample size in a prospective manner. Apra et al. found that exposure of the sinus was a risk factor for rCVST [16]. It is worth mentioning that all patients with cCVST had at least the transverse sinus exposed.

Noteworthily, we found that rCVSTs occurred more frequently when the surgery was performed on the dominant sinus side. Our findings differ from the results of Guazzo et al., who discovered an association between CVSTs and surgery on the side of non-dominant sinus. However, they evaluated only the translabyrinthine approach, which may make direct comparisons inappropriate [18].

### 4.2. Diagnosis and Presentation

A CVST diagnosis is hard to establish, particularly when symptoms are mild. Headaches associated with CVST are unspecific, and their absence cannot rule out CVST [19]. Importantly, patients might have neurologic deficits—or headaches—after retrosigmoid craniotomy, even without CVST. On the other hand, it may be possible that some postoperative neurological deteriorations are in fact caused by CVSTs that are undiagnosed. Still, most CVSTs are clinically silent. The reason for wide spectrum of CVST clinical presentation may be attributed—to some degree—to different venous drainage patterns [20]. The relevant collateral circulations consist of torcular Herophili, contralateral transverse and sigmoid sinuses, the occipital sinus, the vein of Labbe, and the superficial middle cerebral vein. Most studies report CT venography or MR venography as a study of choice for CVSTs [21,22,23].

It is a tenet in our institution that diagnosing a CVST is primarily clinical. We are watchful for alarming headaches in the postoperative period that do not respond to the usual painkillers. In such cases, a contrast enhanced thin-sliced CT scan is an auxiliary diagnostic tool. Yet, the most burning question remains: should only the presence of radiological features of CVST necessitate aggressive pharmacologic treatment?

### 4.3. Medical Treatment

Prospective studies suggest conservative management [14,18]. Guazzo et al. suggested no anticoagulation in incidental rCVSTs [18]. Benjamin et al. chose observation in 23 out of 24 asymptomatic patients with rCVST (the remaining patient was hydrated with IV fluids) [14]. Interestingly, both these studies reported increased CSF leak rates in patients with rCVST, as did Shew et al. in a retrospective cohort study [24]. Orlev et al. reported that most cases resolve without anticoagulation treatment [25]. Kow et al. suggested anticoagulation treatment for symptomatic cases, or those in which rCVST occurs on the dominant sinus side [26]. On the other hand, Moore et al. radically recommended 6 months of anticoagulants in patients with CVST [13]. We suggest aggressive pharmacologic treatment—enoxaparin, mannitol, dexamethasone, IV fluids—only in symptomatic cases. This is because there were no readmissions or any other complications for asymptomatic rCVST cases (which constituted the majority of rCVST sample: 95.1%, 58/61). That said, we believe that asymptomatic rCVST patients should be instructed before discharge about possible cCVST.

### 4.4. Interventional Treatment

Outside of medical treatment, certain cases may require intervention. A large intracerebral hemorrhage may be removed surgically, as described herein. Similarly, endovascular venous thrombectomies have been reported, as unsuccessfully attempted in our fatal cCVST case. The first successful endovascular treatment of CVST after translabyrinthine vestibular schwannoma resection was reported by Manzoor et al. in 2016 [27]. Since then, endovascular treatment of postoperative CVST has not been described in the literature.

### 4.5. Comparison with Other Studies

Appendix A presents case series and case reports describing postoperative CVSTs. Prospective and retrospective studies vary markedly in the presented incidence rates—the former report rates in the range of 32.4% to 38.9% (median 35.7%), while the latter report rates in the range of 0.8% to 22.4% (median 9.1%). Our results are comparable with prospective studies, which may be attributed to the meticulous radiological analysis we performed. Nevertheless, the incidence in clinical practice remains to be evaluated, bearing in mind a broad category of posterior fossa tumors and a narrower category of cerebellopontine angle tumors and specifically vestibular schwannomas. Some authors suggested increased CVST rate after meningioma surgery.

We found that less than 5% of rCVSTs were symptomatic, which is similar to the results of Shew et al., who reported 1 symptomatic patient out of 22 (4.5%) in a study with a similar design to ours [24].

### 4.6. Pediatric Population

This paper focuses exclusively on adult patients. Noteworthily, there is a lack of studies describing CVST in a pediatric population after skull base surgery. Teping et al. described a 7.1% rate of intraoperative sinus bleeding after posterior fossa surgery in the semisitting position in the pediatric population [28]. Petrov et al. reported sinus exposure and injury as potential risk factors for CVST; however, their study included any type of cranial surgery [29].

### 4.7. Approach Selection in CPA Surgery

Currently available reports suggest higher frequencies of CVST after a translabyrinthine approach when compared to using the RSA [13], possibly due to the wide exposure of the sigmoid sinus necessary during such an approach. This remains to be objectively evaluated, and hopefully will be in future prospective studies. It may become an additional factor to consider in choosing the best surgical approach to a given CPA pathology.

### 4.8. Rare Case Reports

A CVST distant to the operative field is a rare event, described in the literature on a case-report basis. There are few case reports in the literature, and one case in our series ended in death [27,30,31,32]. 

### 4.9. Limitations and Future Studies

Though this study has a larger sample size than similar studies (Appendix A), this is still a limiting factor along with its retrospective nature. This may be a reason why, in our study, there were no significant differences in headache occurrences among patients with and without rCVST. This may also be a cause of lack of correlation between the sinus exposure and cCVST. That said, oncologic past medical history was found more frequently among cCVST patients with significance. Finally, we measured only the width of the exposure of each of the sinuses. Additional measurements of the length (or, ideally, the area of surface) of the exposure would improve theoretical accuracy. Unfortunately, in practice, these were impossible to perform reliably. Future prospective studies are needed to evaluate CVST incidence and management, and, in particular, the necessity of prophylactic treatment of asymptomatic patients with rCVST.

## 5. Conclusions

Although it rarely becomes symptomatic, rCVST is a common, underreported consequence of RSA. Surgery on the side of the dominant sinus seems to be related to rCVST. Exposing the sinuses during craniotomy may contribute to rCVST, but it requires more analysis. Otherwise, we did not find any other risk factors associated with either cCVST or rCVST, in particular intraoperative sinus injury. Further, prospective studies are needed to identify risk factors.

## Figures and Tables

**Figure 1 brainsci-13-01039-f001:**
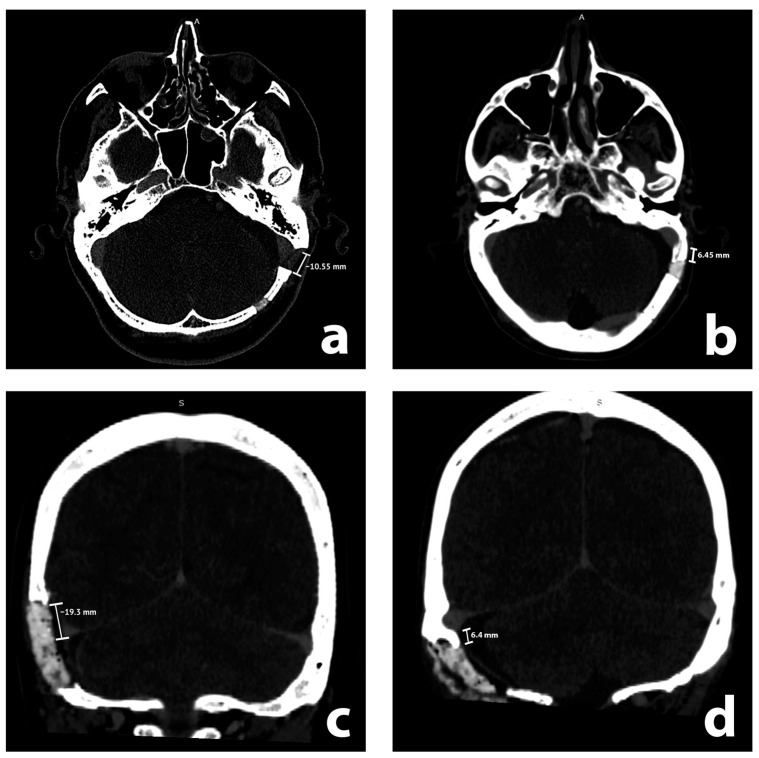
Illustrative radiological axial (**a**,**b**) and coronal (**c**,**d**) contrast enhanced postoperative computed tomography scans with measurements. Sigmoid and transverse sinuses are exposed in (**a**,**c**), while unexposed in (**b**,**d**), respectively.

**Figure 2 brainsci-13-01039-f002:**
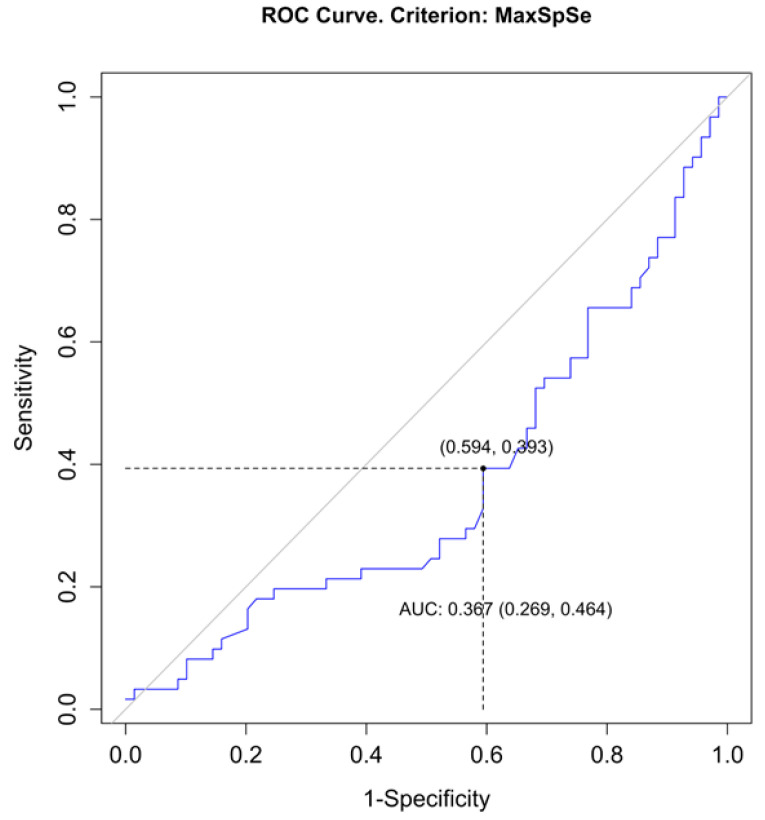
A receiver operating characteristic (ROC) curve analysis shows that exposing a transverse sinus by over 6.55 mm increases the risk of radiologic features of cerebral venous sinus thrombosis (rCVST). ROC curve is shown in blue color, reference line is shown in grey, an optimal cut-point according to the MaxSpSe method is shown in black dashed lines.

**Table 1 brainsci-13-01039-t001:** Radiological analysis. CVST—cerebral venous sinus thrombosis, N/A—not applicable.

Data	Measurement	*p*-Value
Distance to the sigmoid sinus (mm)		N/A
median (range)	−1.75 (−10.55, 7.45)	
1st, 3rd quartile	−5.23, 0.95	
Exposure of the sigmoid sinus		0.01
exposed	79 (60.77%)	
hidden	51 (39.23%)	
Distance to the transverse sinus (mm)		N/A
median (range)	−6.60 (−23.15, 9.2)	
1st, 3rd quartile	−10.13, −2.76	
Exposure of the transverse sinus		<0.0001
exposed	108 (83.1%)	
hidden	22 (16.9%)	
Radiologic features of CVST		0.48
present	61 (46.9%)	
none	69 (53.1%)	
Transverse sinus thrombosis		0.01
present	40 (30.8%)	
absent	21 (16.2%)	
Sigmoid sinus thrombosis		0.1
present	37 (28.5%)	
absent	24 (18.5%)	
Junctions of sinuses thrombosis		0.7
present	32 (24.6%)	
absent	29 (22.3%)	
Intraparenchymal hemorrhage		<0.0001
present	2 (1.5%)	
absent	128 (98.5%)	
Sinus dominance		<0.0001
right	94 (72.3%)	
left	32 (24.6%)	
none	4 (3.1%)	

**Table 2 brainsci-13-01039-t002:** Clinical results of analysis. CVST—cerebral venous sinus thrombosis.

Data	Measurement
Clinical diagnosis of CVST	
present	4 (3.1%)
absent	126 (96.9%)
Intraoperative injuries of the sinuses	
present	6 (4.6%)
absent	124 (95.4%)
Clinical risk factors	
Overweight/obesity	23 (17.7%)
Oncologic past medical history	10 (7.7%)
Deep venous thrombosis	2 (1.5%)
Chronic venous insufficiency	3 (2.3%)
Other	14 (10.8%)
Length of stay (days)	
median (range)	11 (8–48)
1st, 3rd quartile	10, 14
Postoperative headaches refractory to regular medical treatment	
present	12 (9.2%)
absent	115 (88.5%)
not applicable/patient unconscious	1 (0.8%)
no data	2 (1.5%)
Exposure of the transverse sinus	
exposed	108 (83.1%)
hidden	22 (16.9%)

**Table 3 brainsci-13-01039-t003:** Selected analyzed radiological and clinical factors and their correlation with any type of radiological features of cerebral venous sinus thrombosis (rCVST).

Analyzed Factor	Value	rCVST Present	rCVST Absent	*p*-Value
Sex	male	19 (37.3%)	32 (62.7%)	0.076
	female	42 (53.2%)	37 (46.8%)	
Age (years)	median (range)	48 (22–76)	43 (20–82)	0.774
	1st, 3rd quartile	40, 58	33, 57	
Side	right	26 (40.6%)	38 (59.4%)	0.157
	left	35 (53.0%)	31 (47.0%)	
Tumor maximal size (mm)	median (range)	31 (12–50)	32 (9–57)	0.810
	1st, 3rd quartile	24, 39	22, 38	
Tumor volume (cm^3^)	median (range)	11 (0.26–41.89)	11.73 (0.18–59.9)	0.493
	1st, 3rd quartile	3.78, 18.25	3.94, 18.38	
Distance to the sigmoid sinus (quantitative)	median (range)	−2.67 (−9.4–6.2)	−1.6 (−10.6–7.5)	0.04
	1st, 3rd quartile	−5.6, 0	−4.75, 1.95	
Exposure of the sigmoid sinus (qualitative)	exposed	41 (51.9%)	38 (48.1%)	1
	hidden	20 (39.2%)	31 (60.8%)	
Distance to the transverse sinus (quantitative)	median (range)	−7.3 (−21.5–9.2)	−4.9 (−23.15–7.8)	0.028
	1st, 3rd quartile	−11.4, −5.15	−7.95, −2.5	
Exposure of the transverse sinus (qualitative)	exposed	53 (49.1%)	55 (50.9%)	1
	hidden	8 (36.4%)	14 (63.6%)	
Exposure of both sinuses (qualitative)	both exposed	38 (52.1%)	35 (47.9%)	0.46
	one exposed or none	23 (40.4%)	34 (59.6%)	
Exposure of at least one sinus (qualitative)	one or two sinuses exposed	56 (49.1%)	58 (50.9%)	0.180
	none exposed	5 (31.3%)	11 (68.8%)	
Sinus dominance	right	42 (44.7%)	52 (55.3%)	0.46
	left	18 (56.3%)	14 (43.8%)	
	none	1 (25%)	3 (75%)	
Overweight/obesity	present	12 (52.2%)	11 (47.8%)	0.578
	absent	49 (45.8%)	58 (54.2%)	
Oncologic past medical history	present	6 (60%)	4 (40%)	0.514
	absent	55 (45.8%)	65 (54.2%)	
Deep venous thrombosis	present	1 (50%)	1 (50%)	1
	absent	60 (49.6%)	68 (53.1%)	
Chronic venous insufficiency	present	2 (66.7%)	1 (33.3%)	0.600
	absent	59 (46.5%)	68 (53.5%)	
Other clinical risk factors	present	7 (50%)	7 (50%)	0.807
	absent	54 (46.6%)	62 (53.4%)	
Postoperative headaches refractory to regular medical treatment	present	6 (50%)	6 (50%)	0.546
	absent	53 (46.1%)	62 53.9%)	
	not applicable/patient unconscious	1 (100%)	0	
	no data	1 (50%)	1 (50%)	
Surgery on the side of dominant sinus	yes	21 (36.8%)	36 (63.2%)	0.042
	no	40 (54.8%)	33 (45.2%)	
Intraoperative injuries of the sinus	yes	4 (57.1%)	3 (42.9%)	0.705
	no	57 (46.3%)	66 (53.7%)	

## Data Availability

The data presented in this study are available on request from the corresponding author after acceptance of all the co-authors.

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
