# Peer review of "Postoperative Cerebral Venous Sinus Thrombosis Following a Retrosigmoid Craniotomy—A Clinical and Radiological Analysis"

_brainsci, 2023, doi:10.3390/brainsci13071039_

Round 1

Reviewer 1 Report

1-The occurrence of cCVST is very less, which is not correlated well with the exposure to the transeverse or sigmoid sinus. needs explanation.

2-In conclusion nothing specific has been advised for the reader regarding the prevention of CVST , more so for craniotomy design in RSA.

Author Response

Point 1: The occurrence of cCVST is very less, which is not correlated well with the exposure to the transeverse or sigmoid sinus. needs explanation.

Response 1: We thank the Reviewer for this comment. We completely agree with the fact that the correlation between sinus exposure and cCVST is surprising. The reason for that may be – as correctly pointed out by the Reviewer – a low occurrence of cCVST. We would like to point out that this has been addressed in the limitations section of our paper, quote:

“4.9. Limitations and future studies

Though this study has a larger sample size than similar studies (Supplementary Table 3), this is still a limiting factor along with its retrospective nature.”

However, to better address this comment, we have added a following sentence in the above-mentioned fragment:

“4.9. Limitations and future studies

Though this study has a larger sample size than similar studies (Supplementary Table 3), this is still a limiting factor along with its retrospective nature. This may be a reason why in our study there were no significant differences in headache occurrences among patients with and without rCVST. This may also be a cause of lack of correlation between the sinus exposure and cCVST. That said, oncologic past medical history was found more frequently among cCVST patients with significance.”

Point 2: In conclusion nothing specific has been advised for the reader regarding the prevention of CVST , more so for craniotomy design in RSA.

Response 2: Based on our studies, it seems that surgery on the side of the dominant sinus and exposure of the sinuses during the craniotomy may contribute to the postoperative radiological CVST (as per Abstract and Conclusions). This is the reason why, in our practice, as a tenet we do a retrosigmoid craniotomy with minimal to no sinus exposure; accordingly, we would suggest that to everyone. However, we are dedicated to reporting exclusively conclusions that are fully supported by the results of our research. For that reason, our conclusions may seem limited, but they are scientifically accurate.

For us to express this in the conclusions we would have to mark it explicitly as speculative, which we feel is inappropriate.

Reviewer 2 Report

Authors present a retrospective study on 130 patients who underwent elective retrosigmoid craniotomy for various pathologies to investigate incidence of radiological (r) and clinical (c) cerebral sinus venous thrombosis (CSVT).   A striking high number of  rCVST was reported in 46.9% of cases, most often in the transvers sinus/TS (65.6%), and cCVST in 3.1% of cases. Lower exposure of the TS correlated with a lower incidence of rCVST; when surgery was on the side of the dominant sinuses, rCVSTs were more frequent. 

Major drawback is low number of patients and different pathologies and indications for surgery. It is unclear if all of these patients received a postoperative CT angiography, i.e. phlebography - that is the only way to know for sure if there is a CSVT or not, especially taken into consideration that in most of cases one side is dominant. Furthermore, radiological analysis was conducted by two neurosurgeons and two medical students (?) previously instructed (?) by neuroradiologist - why not by two neuroradiologists with similar experience? Please explain. Measurment method of distance from craniotomy to sinus is not very useful - which point of the sinus system? Why only 2D distance? Figure which depicts the measurment, Figure 1. , is suboptimal; please add a better figure. Was image guidance, i.e. navigation or augmented reality used for craniotomy planning? For all four cases of clinical SVT I suggest to include illustrations - preoperative and postoperative imaging as well as if possible intraoperative photo. How was the exposure of the sinus estimated - operative report or postoperative imaging? How was exposure defined - the whole circumference of the sinus, just a bit and if a bit - how much? Include rates of rCSVT in a Table from literature review and compare with your results. What was the overall complication rate? How many patients did use blood thinners or had coagulation disorder? Supplementary Table 2 is not to be found, please add. 

Minor English language editing for improved clarification. 

Author Response

Authors present a retrospective study on 130 patients who underwent elective retrosigmoid craniotomy for various pathologies to investigate incidence of radiological (r) and clinical (c) cerebral sinus venous thrombosis (CSVT).   A striking high number of  rCVST was reported in 46.9% of cases, most often in the transvers sinus/TS (65.6%), and cCVST in 3.1% of cases. Lower exposure of the TS correlated with a lower incidence of rCVST; when surgery was on the side of the dominant sinuses, rCVSTs were more frequent.

Point 1: Major drawback is low number of patients and different pathologies and indications for surgery.

Response 1: We agree with the Reviewer that our study is burdened with a certain degree of limitations, one of which being a low sample size. This has been described in our paper at the end of the Discussion Section under subsection “4.9. Limitations and future studies”. However, we would like to point out that our study has a larger sample size than these already published in the literature (Supplementary Table 3).

In our study we included patients undergoing RSA, which is the reason why there are patients with different pathologies. We disagree that this is a limitation; on the contrary – we think that a wider spectrum of tumors presents a broader and more accurate to the daily practice view on the subject. The indications for the skull base surgery (e.g., observation vs microsurgery vs radiosurgery for a VS) seem irrelevant at this early stage of research in the setting of discussing postoperative, “technical” complications.

Point 2: It is unclear if all of these patients received a postoperative CT angiography, i.e. phlebography - that is the only way to know for sure if there is a CSVT or not, especially taken into consideration that in most of cases one side is dominant.

Response 2: We evaluated routine postoperative contrast–enhanced CT scans (as per “2.1. Study design” subsection). We agree with the Reviewer that this is not a gold standard study – which in that case would be digital subtraction angiography (DSA). Radiological studies that can be used to diagnose CVST are mentioned in “4.2. Diagnosis and presentation” in the Discussion of our manuscript. Please note that we have limited our study only to 2016 and not used earlier studies, as their quality was not good enough for accurate evaluation – as mentioned in subsection “2.1. Study design” (“We could not reconstruct and evaluate studies prior to 2016 because of their insufficient quality.”).

Point 3: Furthermore, radiological analysis was conducted by two neurosurgeons and two medical students (?) previously instructed (?) by neuroradiologist - why not by two neuroradiologists with similar experience? Please explain.

Response 3: The reason for that is the availability of resources. All our measurements were performed on radiological hardware and software, under a senior neuroradiologist direct evaluation and supervision. Additionally, all the measurements were done twice to increase their accuracy.

Point 4: Measurment method of distance from craniotomy to sinus is not very useful - which point of the sinus system? Why only 2D distance?

Response 4: We have described in the Materials and Methods, for each distance we used the closest distance from the edge of craniotomy to the sinuses (subsection 2.3. Radiological analysis). We measured the closest distance to the sigmoid sinus on axial scans, and the closest distance to the transverse sinus on coronal scans.

We explained in the limitations section that measurements of the area of surface of the exposed sinus (in other words, 3D measurements) would be ideal, but these were not unfortunately possible. Additionally, if such measurements were possible, they would have had little practical implications as they probably would not be replicated in everyday practice.

We kindly ask the Reviewer to let us know if the description of the measurement method in the manuscript is clear as it is or if it requires an additional description.

Point 4: Figure which depicts the measurment, Figure 1. , is suboptimal; please add a better figure.

Response 4: We agree with the Reviewer. The reason for that is twofold. Firstly, the image quality in the manuscript file is not the target quality of the figure that we submitted. Additionally, as we mentioned earlier, we used radiologic display stations to perform our evaluation. This is hard to picture in a figure with reduced quality as the Reviewer was evaluating. For these reasons, Figure 1 serves only as an adjunct visualization of a method used in this study and a better-quality image will be available to the publisher.

Point 5: Was image guidance, i.e. navigation or augmented reality used for craniotomy planning?

Response 5: We routinely do not use neuronavigation for the RSA. This was described in the subsection “4.1. Mechanisms and risk factors” of the Discussion (lines 216 – 219, quote: “In our institution, craniotomy technique is focused on minimal (ideally none) expo-sure of the sigmoid and transverse sinuses. To do this, we evaluate preoperative CT scans, use anatomical landmarks and place initial burr hole just below the expected junction of the sinuses.[1] As a rule, we do not use neuronavigation for RSA. [17]”).

Point 6: For all four cases of clinical SVT I suggest to include illustrations - preoperative and postoperative imaging as well as if possible intraoperative photo.

Response 6: We agree with the Reviewer that this would be an interesting addition to the article. We would like to note, however, that the table with description of these patients is already a supplementary file. We think that adding an additional 8 pictures would be excessive and superfluous in this article. For that reason, we would restrain from including additional figures.

Point 7: How was the exposure of the sinus estimated - operative report or postoperative imaging? How was exposure defined - the whole circumference of the sinus, just a bit and if a bit - how much?

Response 7: Exposure of the sinus was estimated with our measurements. In other words, our continuous measurements were translated into qualitative evaluations of sinus exposure. For the nonnegative measurements, the sinus was marked as exposed, and for the others – as not exposed. This is described in the Materials and methods in subsection “2.3. Radiological analysis”. We kindly ask the reviewer to inform us if the description that is already in the manuscript will suffice.

Point 8: Include rates of rCSVT in a Table from literature review and compare with your results.

Response 8: The rates of CVST are included in the Supplementary Table 3 titled “Literature review on the cerebral venous sinus thrombosis after cerebellopontine angle surgery” – the third column from left is “incidence”. At the very bottom of the table there is our study with our rate.

Point 9: What was the overall complication rate? How many patients did use blood thinners or had coagulation disorder?

Response 9: We did not report the overall complication rate, as well as other postoperative data as it is not the aim of the study. All the surgeries were elective, thus all patients on blood thinners had their medications changed to heparin prior to the surgery. This information is not available for us to report. However, we agree with the Reviewer that future prospective studies should include such information.

Point 10: Supplementary Table 2 is not to be found, please add.

Response 10: We believe that this must be a technical mistake as we have double checked that the supplementary file that we have submitted contains the Supplementary Table 2. We kindly ask the Reviewer to inform us if this problem persists and if so, we will contact the Section Managing Editor.

Reviewer 3 Report

The authors present a interesting topic of complication after craniotomy in the posterior fossa.

a thrombosis of transversal or sagittal sinus might be perilous. The diagnostic and the risk factors are not well described yet. The authors present der series and analysed these patients with thrombosis.

The manuscript is well written. The methods ok. The results are conclusive. however, the authors should respond to the following comments and improve their manuscript:

1. please add more informations of the surgical treatment and strategy in case of a sinus injury. 

2. Please add informations about positioning and setting of the patients. Semi-sitting or park bench or posterior? This is important for the understanding of the results

3. please add the following manuscript and discuss the pediatric population:

The semisitting position in pediatric neurosurgery: pearls and pitfalls of a 10-year experience.

Teping F, Linsler S, Zemlin M, Oertel J.J Neurosurg Pediatr. 2021 Oct 1;28(6):724-733. doi: 10.3171/2021.6.PEDS21161.   Do you have results of pediatric patients in your department?   4. Was the research performed with approval of an ethic comittee? And was is a retrospective analysis?

please check again the manuscript by native speaker. There are some minor typing and grammar errors.

Author Response

The authors present a interesting topic of complication after craniotomy in the posterior fossa.

a thrombosis of transversal or sagittal sinus might be perilous. The diagnostic and the risk factors are not well described yet. The authors present der series and analysed these patients with thrombosis.

The manuscript is well written. The methods ok. The results are conclusive. however, the authors should respond to the following comments and improve their manuscript:

Point 1: please add more informations of the surgical treatment and strategy in case of a sinus injury.

Response 1: We would like to thank the Reviewer for this invaluable suggestion. To address that, we have added an appropriate paragraph in the Discussion of our manuscript, subsection “4.1. Mechanisms and risk factors”, quote:

“In case of a sinus injury, our strategy is tailored to the extent of bleeding. Just exposing the sinus, even partially, is followed by keeping it moist and applying a layer of an absorbable hemostat (e.g., Surigcel ®) on the surface or just patties. Minor bleedings are dealt with similarly. More serious bleedings are most often successfully stopped with a fibrin sealant patch (e.g., TachoSil ®). A direct sinus injury is sutured. “

Point 2: Please add informations about positioning and setting of the patients. Semi-sitting or park bench or posterior? This is important for the understanding of the results

Response 2: We thank the reviewer for this comment, we agree that this is an important detail that was missing from our paper. We routinely perform retrosigmoid craniotomy in supine position with a head locked and turned in skull clamp. This information was added to the Discussion in subsection “4.1. Mechanisms and risk factors”.

Point 3: please add the following manuscript and discuss the pediatric population:

The semisitting position in pediatric neurosurgery: pearls and pitfalls of a 10-year experience.

Teping F, Linsler S, Zemlin M, Oertel J.

J Neurosurg Pediatr. 2021 Oct 1;28(6):724-733. doi: 10.3171/2021.6.PEDS21161.

Do you have results of pediatric patients in your department?

Response 3: We agree with the Reviewer that a pediatric population may be an interesting subgroup, potentially with different occurrences, risk factors and presentation. Unfortunately, we do not treat pediatric patients and have no data nor experience on that.

We have added a subsection to our discussion regarding pediatric population (4.6. Pediatric population”). We have also broadened our bibliography with the article suggested by the Reviewer (ref. no XXX). We kindly ask the Reviewer for an opinion if that will suffice to explore this subject.

Point 4: Was the research performed with approval of an ethic comittee? And was is a retrospective analysis?

Response 4: We salute the Reviewer for the diligence that is verbalized with this question. This study was approved by the local institutional review board (The Bioethics Committee of the Medical University of Warsaw). This information is provided at the end of the manuscript (just prior to the References). Regarding the design of the study, this was a retrospective analysis, which is also described in the Material and Methods section as well as in the subsection “4.8. Limitations and future studies”. We kindly ask the Reviewer to inform us if the information that is already in the manuscript is sufficient or if the Reviewer suggests that this needs to be emphasized more.

Round 2

Reviewer 2 Report

The authors have responded to some of the reviewer remarks. However, my humble opinion remains that an additional figure is needed and that the complication rate and clinical and radiological outcome should be included. 

Acceptable. 

Author Response

Response to Reviewer 2 Comments

Point 1: The authors have responded to some of the reviewer remarks. However, my humble opinion remains that an additional figure is needed

Response 1: We would like to meet the Reviewer’s requests, for that reason an additional figure was added as a supplement.

Point 2: and that the complication rate and clinical and radiological outcome should be included.

Response 2: We would like to make the point that this was not the aim of our study. This is important because our actions are limited by the IRB approval. To get the most up-to-date follow-up, we would need to do e.g., phone interviews with all the study participants. Unfortunately, this was not allowed by our IRB and would not be in accordance with the Declaration of Helsinki.

Additionally, we would also like to point out that our institution has already published several papers in which our results (including mentioned by the Reviewer complication rate, clinical and radiological outcomes) have already been extensively discussed (1-7). We would not like to repeat and publish the same material – it is unnecessary and unethical.

References:

  1. Kunert, P., Dziedzic, T., Nowak, A., Czernicki, T., & Marchel, A. (2016). Surgery for sporadic vestibular schwannoma. Part I: General outcome and risk of tumor recurrence. Neurologia i neurochirurgia polska, 50(2), 83–89. https://doi.org/10.1016/j.pjnns.2016.01.001
  2. Kunert, P., Dziedzic, T., Czernicki, T., Nowak, A., & Marchel, A. (2016). Surgery for sporadic vestibular schwannoma. Part II. Complications (not related to facial and auditory nerves). Neurologia i neurochirurgia polska, 50(2), 90–97. https://doi.org/10.1016/j.pjnns.2016.01.002
  3. Kunert, P., Dziedzic, T., Podgórska, A., Czernicki, T., Nowak, A., & Marchel, A. (2015). Surgery for sporadic vestibular schwannoma. Part III: Facial and auditory nerve function. Neurologia i neurochirurgia polska, 49(6), 373–380. https://doi.org/10.1016/j.pjnns.2015.08.008
  4. Kunert, P., Dziedzic, T., Podgórska, A., Nowak, A., Czernicki, T., & Marchel, A. (2016). Surgery for sporadic vestibular schwannoma. Part IV. Predictive factors influencing facial nerve function after surgery. Neurologia i neurochirurgia polska, 50(1), 36–44. https://doi.org/10.1016/j.pjnns.2015.11.006
  5. Nowak, A., Dziedzic, T., Czernicki, T., Kunert, P., Morawski, K., Niemczyk, K., & Marchel, A. (2015). Strategy for the surgical treatment of vestibular schwannomas in patients with neurofibromatosis type 2. Neurologia i neurochirurgia polska, 49(5), 295–301. https://doi.org/10.1016/j.pjnns.2015.06.008
  6. Kunert, P., Smolarek, B., & Marchel, A. (2011). Facial nerve damage following surgery for cerebellopontine angle tumours. Prevention and comprehensive treatment. Neurologia i neurochirurgia polska, 45(5), 480–488. https://doi.org/10.1016/s0028-3843(14)60317-0
  7. Przepiórka, Ł., Kunert, P., Rutkowska, W., Dziedzic, T., & Marchel, A. (2020). Surgery After Surgery for Vestibular Schwannoma: A Case Series. Frontiers in oncology, 10, 588260. https://doi.org/10.3389/fonc.2020.588260

Reviewer 3 Report

accept as it is now

Author Response

We thank the Reviewer for the suggestions and comments. We believe that our manuscript has been improved.